# Assessing electrocardiogram changes after ischemic stroke with artificial intelligence

**Ziqiang Zeng[1,2☯], Qixuan Wang[3☯], Yingjing Yu[1,2], Yichu Zhang[4], Qi Chen[4], Weiming Lou[5], Yuting Wang[1,2], Lingyu Yan[1,2], Zujue Cheng[6,7], Lijun Xu[8], Yingping Yi[9], Guangqin Fan[1,2], Libin Deng[1,2,10]***

**1** Jiangxi Provincial Key Laboratory of Preventive Medicine, Nanchang University, Nanchang, P.R. China, **2** School of Public Health, Nanchang University, Nanchang, China, **3** Queen Mary School, Medical College of Nanchang University, Nanchang, China, **4** Department of Cardiovascular Medicine, The Second Affiliated Hospital of Nanchang University, Nanchang, China, **5** Institute of Translational Medicine, Nanchang University, Nanchang, China, **6** Department of Neurosurgery, The Second Affiliated Hospital of Nanchang University, Nanchang, China, **7** Institute of Neuroscience, Nanchang University, Nanchang, China, **8** Department of Neurology, The Second Affiliated Hospital of Nanchang University, Nanchang, China, **9** Department of Medical Big Data Center, The Second Affiliated Hospital of Nanchang University, Nanchang, China, **10** The Institute of Periodontal Disease, Nanchang University, Nanchang, China

☯ These authors contributed equally to this work.
* lbdeng@ncu.edu.cn

**Data Availability Statement:** All original data we used is available in the figshare database (DOI: https://doi.org/10.6084/m9.figshare.21523527.v1) for reproduction. The request for source code may be considered by the principal investigator.

## Abstract

### Objective

Ischemic stroke (IS) with subsequent cerebrocardiac syndrome (CCS) has a poor prognosis. We aimed to investigate electrocardiogram (ECG) changes after IS with artificial intelligence (AI).

### Methods

We collected ECGs from a healthy population and patients with IS, and then analyzed participant demographics and ECG parameters to identify abnormal features in post-IS ECGs. Next, we trained the convolutional neural network (CNN), random forest (RF) and support vector machine (SVM) models to automatically detect the changes in the ECGs; Additionally, We compared the CNN scores of good prognosis (mRS $\leq$ 2) and poor prognosis (mRS > 2) to assess the prognostic value of CNN model. Finally, we used gradient class activation map (Grad-CAM) to localize the key abnormalities.

### Results

Among the 3506 ECGs of the IS patients, 2764 ECGs (78.84%) led to an abnormal diagnosis. Then we divided ECGs in the primary cohort into three groups, normal ECGs (N-Ns), abnormal ECGs after the first ischemic stroke (A-ISs), and normal ECGs after the first ischemic stroke (N-ISs). Basic demographic and ECG parameter analyses showed that heart rate, QT interval, and P-R interval were significantly different between 673 N-ISs and 3546 N-Ns ($p < 0.05$). The CNN has the best performance among the three models in distinguishing A-ISs and N-Ns (AUC: 0.88, 95%CI = 0.86–0.90). The prediction scores of the A-ISs and N-ISs obtained from the all three models are statistically different from the N-Ns ($p <$

**Funding:** This work was supported by funding from Hospital-level project of the Second Affiliated Hospital of Nanchang University (No. 2021efyC02); Jiangxi Provincial Science and Technology Department Project (No. 20208031); Major Projects of Institute of Chinese Materia Medica China Academy of Chinese Medical Sciences (No. ZZ15-WT-04); Jiangxi Provincial Education Department Project (No. GJJ200133).

**Competing interests:** The authors have declared that no competing interests exist.

0.001). Futhermore, the CNN scores of the two groups (mRS > 2 and mRS $\leq$ 2) were significantly different ($p < 0.05$). Finally, Grad-CAM revealed that the V4 lead may harbor the highest probability of abnormality.

## Conclusion

Our study showed that a high proportion of post-IS ECGs harbored abnormal changes. Our CNN model can systematically assess anomalies in and prognosticate post-IS ECGs.

## Introduction

Stroke is a common cerebrovascular disease, with 10.3 million incidents annually [1], and accounts for 10% of total deaths and 5% of all disability-adjusted life years (DALYs) globally [2]. Ischemic stroke (IS) is a typical brain injury; patients with recent IS events often have cardiac complications, a condition defined as cerebrocardiac syndrome (CCS). CCS resulting from IS contributes to a poor prognosis and is the second leading cause of death within the following few weeks after the event [3, 4]. Recent research has indicated that IS can cause impaired cardiac function even in the absence of related risk factors (such as hypertension, diabetes, hypercholesterolemia) or preexisting heart disease [3, 5, 6]. Some studies have found that acute coronary syndrome, heart failure, and cardiac arrhythmia occur predominantly within the first 3 days after the attack [7]. The underlying reasons may involve the mutual feedback signals linking the two organ systems; stroke induces imbalanced cerebral autoregulation, uncoupling the neurovascular system and forcing cerebral blood flow to rely directly on cardiac function [3, 8, 9]. This may lead to a suddenly increased burden on the heart and thus result in several complications.

Electrocardiograms (ECGs) and nonspecific cardiac blood biomarkers are commonly used for observing cardiac changes after ischemic stroke [4]. Since ECGs are noninvasive examinations, they might be an optimal choice for poststroke monitoring. Changes in repolarization are the most frequently observed ECG changes in the early post stroke phase. Abnormalities include prolonged heart rate-corrected QT (QTc) time in 20–65% of patients, ST-segment changes in 15–25% of patients, and a cerebral T wave in 2–18% of patients (inverted T waves with altered amplitude and width) [4, 10–12]. Additionally, a long QTc can serve as a predictor of cardiac events after IS attack [7]. From these studies, ECG changes including a prolonged QTc and ST depression are frequently observed in post-IS patients with cardiac complications. However, the results of these studies harbor a certain degree of heterogenicity. There is a need for a standard system for evaluating ECG changes and prognoses in the post-IS population.

Artificial intelligence (AI) has shown great potential in identifying abnormal changes in ECGs that are not previously noticed or that require more accurate identification. It has been reported that AI-based ECG analysis can be used to identify left ventricular systolic dysfunction in patients with dyspnea, atrial fibrillation, and concealed long QT syndrome [13–15]. AI applications in cardiology provide a tool for relieving cardiologists from analyzing ever-increasing amounts of data and improve effectiveness [16]. In addition, they can help overcome the limitations of interobserver variability, allowing arrival at a diagnostic consensus [17].

In this study, we aimed to develop an AI-ECG approach to analyze the changes in the ECGs of post-IS patients to explain their poor prognoses. Our research provides indications for better medical attention and therapeutic allocation to prevent exacerbation.

## Materials and methods

### Study population

We performed a retrospective analysis of patients with a first IS attack admitted between 1 Sep 2017 and 31 May 2020 to The Second Affiliated Hospital of Nanchang University. IS was diagnosed following a general and neurological examination, CT, blood tests, and ECG. Patients with previous stroke history, use of drugs influencing the ECG pattern, heart diseases (cardiac arrhythmia, myocardial infarction history), and heart-related disease (i.e., electrolyte imbalance) were excluded. The ECGs of IS patients were collected within one hour of admission. Normal ECGs were collected from a healthy population (without IS or heart diseases) from the physical examination center of the hospital. We only collected one ECG from each participant. The sexes, ages, and ECGs of all participants were included. All 5-second, standard 12-lead ECGs were collected at a 25 mm/s paper speed and 1 mV/10 mm voltage setting (FX-7042 Cardimax ECG machine). The ECGs were initially interpreted by experienced cardiologists with the title of deputy director or above, all having over 10 years of working experience. Patients with ECGs with no continuous P-QRS-T waves in each lead and incomplete related clinical data were excluded. The modified Rankin Scale (mRS) is a commonly used scale for measuring neurological disability and predicting the clinical outcome after a stroke attack. We used the discharge mRS scores to assess the functional outcome of post-IS patients, providing predictions about the clinical endpoint. An mRS score > 2 refers to a poor outcome (severe disability), and a score ≤ 2 refers to a good outcome (no or slight disability). The study was approved by the Second Affiliated Hospital of Nanchang University Medical Research Ethics Committee, and written informed consent was obtained from all participants.

### ECG parameter extraction

All ECGs we collected were divided into primary and validation cohorts. The primary cohort included 3 datasets: a normal ECG (N-N) dataset, a normal ECG after the first ischemic stroke (N-IS) dataset, and an abnormal ECG after the first ischemic stroke (A-IS) dataset. The ECG parameters were generated by the FX-7042 Cardimax ECG machine and recorded in the electronic medical record. Seven ECG parameters, including heart rate, P wave, P-R interval, QRS wave, QT interval, QTc interval, and RV5+SV1 value, were extracted from the electronic medical record and checked manually.

### Convolutional neural network construction

The A-IS dataset includes defined abnormal changes, while the N-N dataset is the completely normal ECG collected from healthy individuals. We aimed to use the N-N and A-IS datasets to train a convolutional neural network (CNN) model that can identify abnormal changes on post-IS ECGs. Then we applied the trained CNN model to identify if the N-IS dataset harbored some abnormal changes that are easily be ignored in routine clinical practice.

We used a supervised CNN model for intrinsic feature extraction from the ECG images. We input the labeled ECGs (the whole image) categorized into eithor A-IS or N-N, to train a model that could identify abnormal ECGs. All ECG images were preprocessed before model development. We cropped the images to 2955×1680 pixels, removed all unnecessary text, and retained only the image information. We used resize and normalize functions to preprocess the ECG (in Python v3.6). In this process, we eliminate errors that may be caused by background pixels and avoid possible confounding factors. The images were resized to 400×400 pixels and converted to single-channel grayscale images. All images were normalized and

standardized. The proportions of ECGs used for training, validation, and testing were 64%, 16%, and 20%, respectively.

In our study, we adjusted the hyperparameters and structure of the convolutional neural network according to the working speed and performance of the model, based on which we chose the optimal hyperparameters and structure for the model. The CNN structure consisted of seven layers: two convolution layers, two max-pooling layers, and three fully connected layers. The convolution layer was used for feature extraction to generate an abstract feature map of the raw image. Padded convolution was adopted to keep the input and output the same size. The max-pooling layers were used to remove the surrounding redundant information and reduce the scale of parameters and the amount of calculation. The fully connected layer used L2 regularization and dropout to prevent overfitting and assigned weights to different features. The network input was 400×400-pixel images. We used stochastic gradient descent (SGD) during the training process to obtain the best fit model parameter between the predicted output and actual output more efficiently. The validation dataset was used to perform hyperparameter optimization on all parameters, and the test dataset was used to finally assess model power. The optimal learning rate was adjusted to 0.001, and the batch size was 64. The rectified linear unit (ReLU) activation function was adopted to establish the model. In the test datasets, a predicted score from 0 to 1 was calculated for each image, indicating the probability of the images being classified as N-N or A-IS. Additionally, we tested the performance of the CNN model with the validation cohort. The pipeline of CNN development is shown in Fig 1A and 1B. The network structure and hyperparameters of the CNN model are shown in S1 Table.

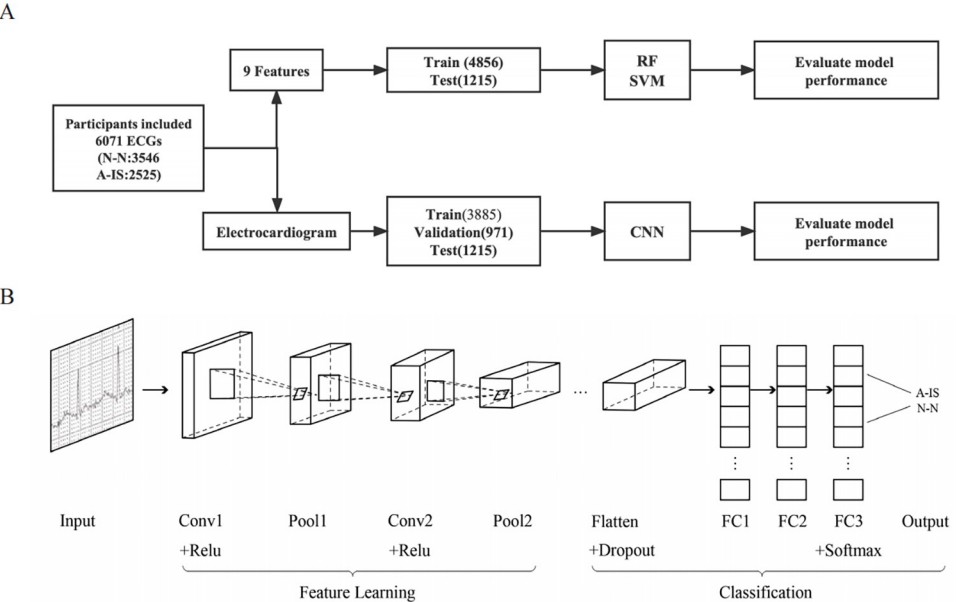

**Fig 1. Pipeline of AI model development and CNN structure.** (A) Overview of the pipeline of AI model construction. A total of 6071 ECGs from the primary cohort were included, and we trained the ML models with the following demographics and ECG parameters: sex, age, heart rate (bpm), P wave (ms), P-R interval (ms), QRS wave (ms), QT interval (ms), QTc interval (ms), and RV5+SV1 (mV). The CNN model was developed using the training and validation datasets as input. After building the model, the test dataset was used to evaluate the final model performance. (B) The structure of the CNN model, including 2 convolution layers (Conv), 2 pooling layers, and 3 fully connected layers (FC). Note: A-IS indicates abnormal ECG after the first ischemic stroke; N-N, normal ECG; RF, random forest; SVM, support vector machine; CNN, convolutional neural network; Conv, convolution layers; Relu, rectified linear unit; FC, fully connected layer.

## Machine learning

The aim of machine learning in this study was to use ECG parameters and demographics (sex and age) of patients in the N-N and A-IS datasets to develop models that could distinguish the two. We developed two supervised ML models: support vector machine (SVM) and random forest (RF), which are based on feature selection. Unlike the CNN which directly takes image inputs, these machine learning models used extracted ECG parameters to perform model training and classification. The weight of each parameter in model development can also be shown as an importance value, allowing some degree of model interpretation by indicating of which parameters are the most important indicators. The ECG parameters included heart rate (bmp), duration of P wave (ms), P-R interval (ms), QRS wave (ms), QT interval (ms), and QTc interval (ms). The variables we used to develop the machine models are shown in Table 1. The data were divided into 2 groups, training, and testing, using the splitting protocol shown in Fig 1A. We used 80% of the ECGs to train the model and 20% for testing. Repeated ten-fold cross-validation was adopted for model training, and the testing dataset was used to assess model discriminability. The scikitlearn library (available in Python v3.6) was used to establish the ML models.

## Gradient class activation map construction

We used a gradient class activation map (Grad-CAM) to reconstruct the image and localize the intensive areas of feature extraction. This method can identify the image features that the CNN used to perform the decision-making process, enabling us to understand what predictive features were determined by the trained model [18]. Here, we used Grad-CAM to identify key abnormal features in the A-IS dataset. Regions with high feature extraction intensity were assumed to be the key points in anomaly feature identification.

## Statistical analysis

The discrimination power of all the models was assessed by calculating the area under the receiver operating characteristic (ROC) curve (AUC). The maximum Youden index derived from the ROC curves was selected as the optimal cutoff value for classifying the ECG based on the model prediction scores. The normality of the data was assessed with the Shapiro–Wilk normality test, and then we used the Wilcoxon rank-sum test on the seven ECG parameters and the prediction scores assessed by the three models between the N-N and N-IS datasets and

**Table 1. Demographics and ECG parameters of participants from the primary cohort and validation cohort.**

|  | Primary cohort (n = 6744) | | | Validation cohort (n = 616) |
| --- | --- | --- | --- | --- |
|  | A-IS dataset (n = 2525) | N-IS dataset (n = 673) | N-N dataset (n = 3546) |  |
| Female sex, n (%) | 1065 (42.18) | 274 (40.71) | 1498 (42.24) | 252 (40.91) |
| Age, mean (SD), y | 67.16 (11.50) | 62.15 (11.29) | 65.85 (12.65) | 66.13 (12.15) |
| Heart rate, mean (SD), bpm | 78 (23) | 74 (10) | 75 (10) | 77 (20) |
| P wave, mean (SD), ms | 88.48 (36.46) | 103.07 (13.19) | 102.80 (12.18) | 93.24 (30.84) |
| P-R interval, mean (SD), ms | 142.13 (61.68) | 158.85 (18.65) | 156.26 (19.85) | 145.45 (50.45) |
| QRS wave, mean (SD), ms | 91.04 (19.34) | 88.80 (7.38) | 88.37 (7.36) | 89.10 (10.04) |
| QT interval, mean (SD), ms | 382.65 (68.32) | 383.96 (25.91) | 380.45 (25.38) | 381.91 (43.52) |
| QTc interval, mean (SD), ms | 426.94 (61.98) | 425.92 (22.92) | 424.59 (22.61) | 427.75 (33.79) |
| RV5+SV1, mean (SD), mV | 2.48 (1.06) | 2.32 (0.62) | 2.35 (0.61) | 2.45 (0.79) |

Note: A-IS indicates abnormal ECG after the first ischemic stroke; N-IS, normal ECG after the first ischemic stroke; N-N, normal ECG; bpm, beats per minute.

between the N-N and A-IS datasets according to their distribution features. Differences in scores between mRS categories assessed by the CNN model were also assessed by the Wilcoxon rank-sum test. We used the matplotib library in Python to draw the ROC curves and other figures. All model development and statistical analyses were performed in Python (v3.6) and MedCalc (v18.11.3). Ziqiang Zeng has full access to all data in this study and takes responsibility for its integrity and analysis.

## Ethics statement

The study was approved by the Second Affiliated Hospital of Nanchang University Medical Research Ethics Committee.

## Results

### Demographics and ECG parameter analysis

In total, we collected 7,052 ECGs from The Second Affiliated Hospital of Nanchang University between 1 September 2017 and 31 May 2020. A total of 4,086 males and 2,966 females (42.06%) >20 years old were included in the research, and the average age was 66.00 ± 12.17 years. Additionally, we included the ECGs of 3546 healthy individuals matched by age and sex screened from other departments. In the primary cohort, we included 3,546 N-Ns, 673 N-ISs, and 2,525 A-ISs. In the validation cohort, we included 308 ECGs from poststroke patients with mRS score annotations (239 A-ISs, 69 N-ISs) and 308 N-Ns (randomly picked from the primary cohort).

Among all the ECGs of IS patients, 2764 ECGs (78.84%) were considered abnormal. T-wave and ST-T–segment changes were the most common ECG changes, accounting for 16.71% and 14.09%, respectively. Atrial fibrillation accounted for 8.76% (Table 2). Demographics and ECG parameters analyses of the 2 cohorts involved in the study are shown in Table 1. Statistical analysis between the N-IS and N-N datasets showed that heart rate (HR, bpm), P-R interval (ms), and Q-T interval (ms) were different ($p < 0.05$, Wilcoxon rank-sum test), as shown in Fig 2. The age difference between the N-N and A-IS groups was not statistically significant. This finding shows that some changes were present in ECGs that were initially

**Table 2. ECG patterns of diagnosed ischemic stroke patients.**

| ECG patterns of IS patients | Amount | Proportion |
|---|---|---|
| Normal | 742 | 21.16% |
| T wave change | 586 | 16.71% |
| ST-T segment changes | 494 | 14.09% |
| Sinus bradycardia | 451 | 12.86% |
| Left ventricular high voltage | 425 | 12.12% |
| Atrial fibrillation | 307 | 8.76% |
| Counterclockwise rotation | 235 | 6.70% |
| Abnormal Q wave | 224 | 6.39% |
| Atrial premature beats | 199 | 5.68% |
| Clockwise rotation | 180 | 5.13% |
| Sinus tachycardia | 177 | 5.05% |
| Ventricular premature beat | 126 | 3.59% |

Note: The total number of IS patients was 3506. Except for patients with normal ECG, most patients have two or more diagnosis results. Proportion of one ECG pattern = number of patients with this ECG pattern/total number of patients with IS. This table only records the ECG patterns that occured more than 100 times.

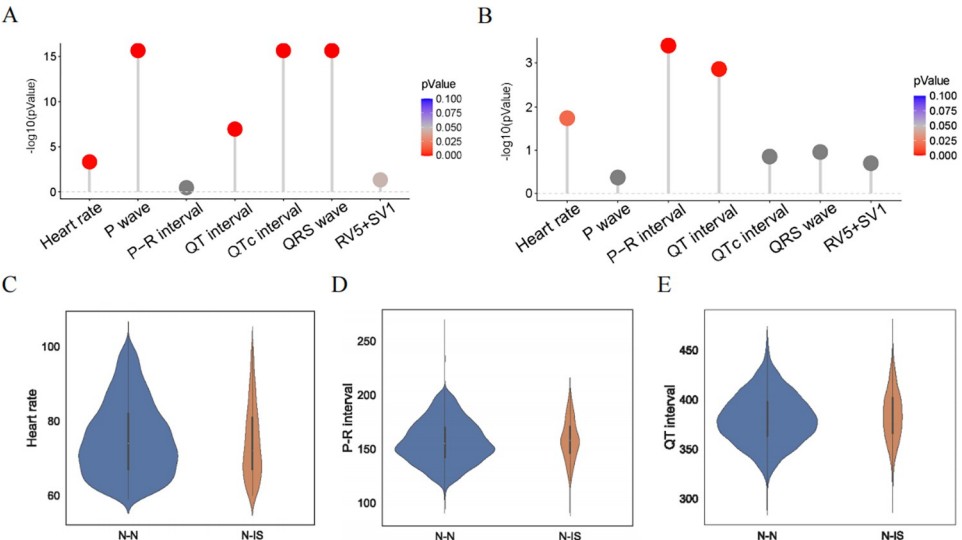

**Fig 2. Analysis of participant ECG parameters.** (A) *P* values for the seven ECG parameters between the A-IS and N-N datasets (Wilcoxon rank-sum test). (B) *P* values for the seven ECG parameters between the N-IS and N-N datasets (Wilcoxon rank-sum test). (C) Heart rate (bpm) distribution of the N-N and N-IS datasets. (D) P-R interval (ms) distribution of the N-N and N-IS datasets. (E) QT interval (ms) distribution of the N-N and N-IS datasets.

classified as normal in post-IS patients. The exact *p* values are listed in S2 Table. We verified that the A-ISs and N-ISs had more changes than the N-Ns.

## Performance of the ML and CNN models

Two ML models and one CNN model were established to obtain a standard for indentifying unusual changes in ECGs from the A-IS dataset. The RF and SVM models used a combination of ECG-related variables derived from each participant (HR, P wave, P-R interval, QRS wave, QT interval, QTc interval, and RV5+SV1 value) as well as sex and age. The CNN model took the image of the ECG as direct input and served as an image classifier. The RF and SVM models had AUCs of 0.83 and 0.82, respectively (Fig 3A, S3 Table). As shown in Fig 3B and 3C, HR and the QR interval were the most important variables in the RF and SVM models, respectively. The CNN model exhibited the best performance among the three models, with an accuracy of 0.82 and AUC of 0.88 (95% CI = 0.86–0.90) (Fig 3A). In the testing dataset, 431 ECGs were considered to be abnormal by the CNN model, and 784 ECGs were considered to be normal (recall = 0.71; precision = 0.84; F1 score = 0.77; Matthews correlation coefficient (MCC) = 0.63, shown in S3 Table). The confusion matrix of the three AI models is shown in S1 Fig.

In addition, we compared the prediction scores of the N-N, A-IS, and N-IS datasets obtained from the three models. The Wilcoxon rank-sum test indicated a significant difference not only between the N-N and A-IS datasets but also between the N-N and N-IS datasets (Fig 3D and S2 Fig). The ECGs with model prediction scores higher than the optimal cutoff value were identified as abnormal ECGs. In the N-IS dataset, 145 ECGs (21.55%) were considered to be abnormal by the CNN model, 54 (8.02%) by the RF model, and 84 (12.48%) by the SVM model. The CNN model identified the most abnormal ECGs in the N-IS dataset.

## CNN model performance in the validation cohort

The CNN model was the best-performing model among the three models, demonstrating the best capacity to identify abnormalities (in the N-IS dataset). We then used it to identify the

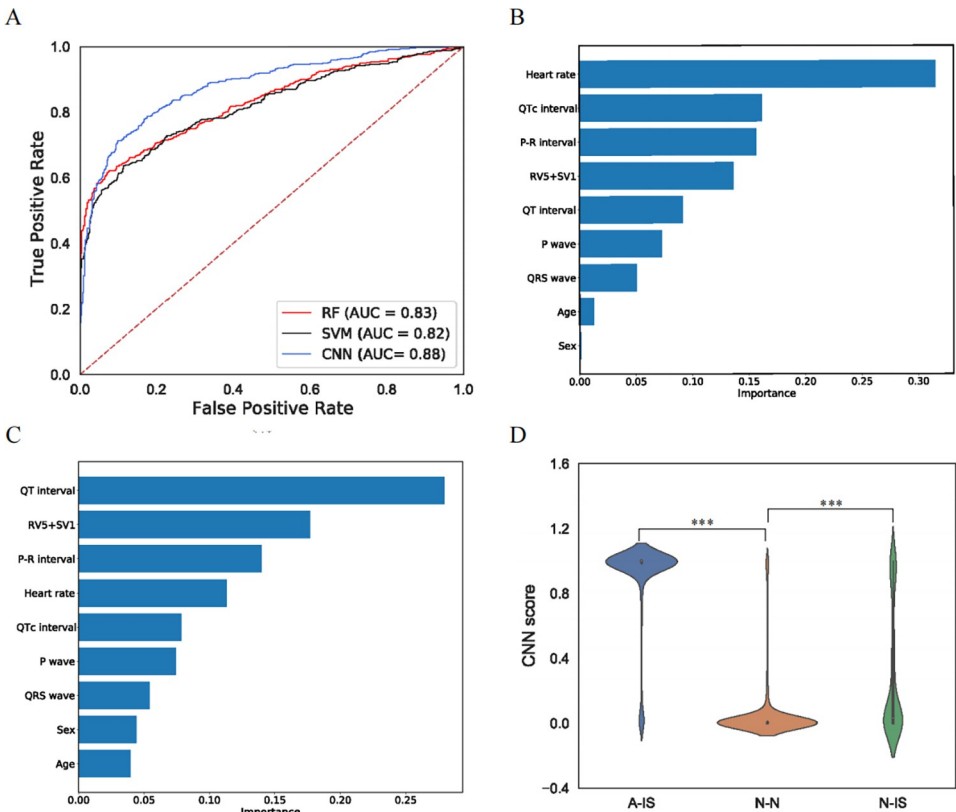

**Fig 3. Performance of the ML and CNN models.** (A) Comparison of receiver operating characteristic (ROC) curves of the CNN model and 2 ML models (RF and SVM). The area under the ROC curve (AUC) of the CNN model (0.88, 95% CI = 0.86–0.90) was higher than that of the RF model (0.83, 95% CI = 0.81–0.85) and SVM model (0.82, 95% CI = 0.80–0.84), indicating the best performance in classification. (B) Variables used to build the RF model are ranked according to their importance. Heart rate is ranked as the most important variable for classification. (C) In the SVM model, the QT interval is ranked as the most important variable for classification among the demographics and ECG parameters. (D) The score distribution of the N-N, N-IS and A-IS datasets was evaluated by the CNN model. The scores of the A-IS dataset are concentrated around 1, while the scores of the N-N dataset are concentrated near 0. The scores of the N-IS dataset do not show an overt central tendency. The scores between the N-N and N-IS datasets are significantly different ($p < 0.001$, Wilcoxon rank-sum test), as are the scores between the A-IS and N-IS datasets. ($p < 0.001$, Wilcoxon rank-sum test). Note: $p < 0.05$; *, $< 0.01$, **; $< 0.001$,***.

proportion of abnormal ECGs in the validation cohort for further confirmation. The accuracy of the CNN model was 0.89. Importantly, 47.83% of the N-IS prediction scores were higher than the optimal CNN model cutoff value and were identified to be abnormal (Fig 4A). This further confirmed that there might be some concealed abnormal features in N-ISs that are present in A-ISs.

To investigate the relationship between ECG changes and prognosis, we assessed the correlation between the prediction scores of the CNN model and mRS scores. As shown in Fig 4B, the scores of the two groups (mRS > 2 and mRS ≤ 2) were significantly different (Wilcoxon rank-sum test, $p < 0.05$). Patients with high CNN scores also had high mRS scores, indicating that the CNN score could reflect the prognosis of post-IS patients.

## Grad-CAM for feature visualization

The reconstructed feature maps enabled us to visualize the intensity of feature extraction in the ECG image and, in turn, detect anomaly key points. As shown in Fig 5A, the example

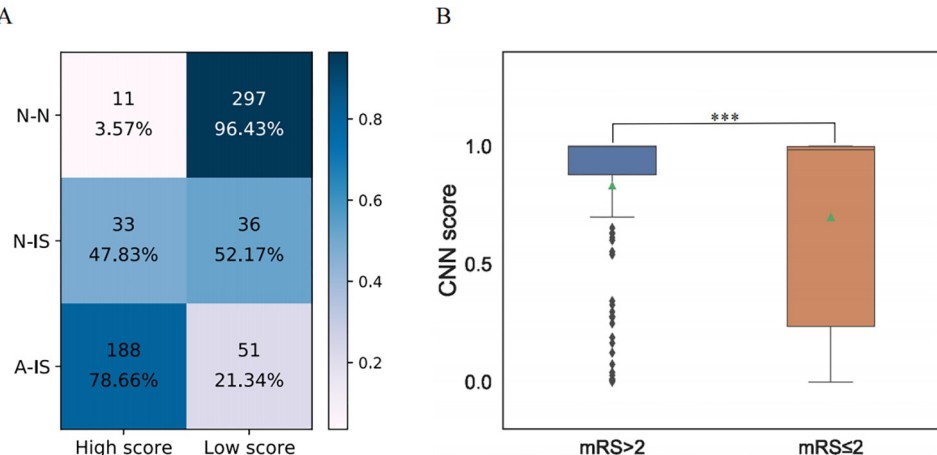

**Fig 4. CNN model performance in the validation cohort and in poststroke disability assessment.** (A) The confusion matrix shows the performance of the CNN with the validation cohort. (B) The CNN score distribution of the ECGs with prognostic annotation according to their mRS scores ($> 2$ poor, $\leq 2$ good); and the scores of the two groups are significantly different ($p < 0.001$). Note: $p < 0.05$, *; $< 0.01$, **; $< 0.001$,***.

heatmap highlights the intensity of feature extraction, reflecting the degree of abnormality in various regions with respect to the N-N image. Regions with a value greater than 0.8 were assumed to be the key points in anomaly feature identification. A total of 360 A-ISs (correctly identified by the CNN model) were evaluated by Grad-CAM, and we counted the number of key points distributed in different leads. In all, 636 key points were identified, with the V3, V4, and V5 leads harboring most of them. The V4 lead harbored 151 key points, indicating that it serves as the most important factor in abnormal feature identification (Fig 5B).

## Discussion

The ECG reflects the electrical activity of the heart, providing a plethora of information on primary heart and heart-related diseases. In addition, it is easy to perform, noninvasive, and highly reproducible, making it the gold standard for heart disease monitoring and essential clinical decision-making, for example, for cardiac arrhythmias [19]. Acute ischemic stroke is often related to severe CCS, which can be reflected by ECGs to some extent. Previous studies

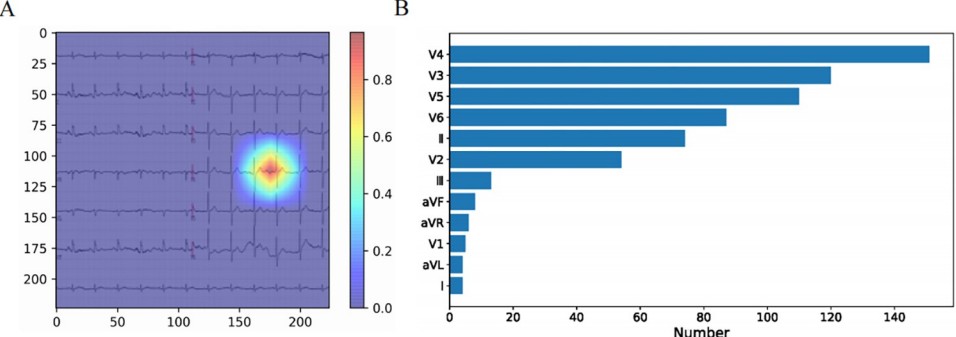

**Fig 5. Grad-CAM feature visualization.** (A) The example heatmap highlights the intensity of feature extraction from the original ECG, which reflects degrees of abnormality in various regions with respect to the N-N image. Regions with values greater than 0.8 are assumed to be key points in anomaly feature identification. (B) Among all important regions, the V3, V4, and V5 leads harbor most of the key points, and the V4 lead in particular serves as the most important factor in abnormal feature identification.

have analyzed the abnormal changes in post-IS ECGs and found that prolonged QTc (indicating that the heart's electrical system takes a longer time to recharge between beats than usual) and ST depression (the ST segment is abnormally below the baseline on ECG) are the most frequently observed abnormalities. However, the proportions of abnormal changes in each ECG feature differed notably among these studies, and no consensus has yet been reached regarding what constitutes an abnormal pattern. The small sample size may partially explain the heterogeneity among these studies. We observed that T-wave and ST-T–segment changes were the most common IS ECG changes, accounting for 16.71% and 14.09% of all cases, respectively. Our observations confirm previous studies indicating that ECGs do change after an IS attack, with T-wave and ST-T–segment changes being the most common features of post-IS ECGs. Atrial fibrillation is a risk factor for stroke occurrence and is related to poor stroke prognosis [20]. We also observed that atrial fibrillation accounts for a certain proportion of abnormal ECG changes. which conforms with previous studies. In this study, we analyzed abnormal ECG patterns in IS patients with a larger sample size, providing relatively more comprehensive and reliable results.

It has been widely accepted that AI can discern subtle differences that human eyes cannot through big data analysis. It is crucial to identify different patterns of ECG abnormalities that might serve as hallmarks of multiple diseases. AI applications in ECG analysis have been extensively reported, especially CNNs, which have excellent performance in ECG analysis [21]. Convolution enables the network to remove unnecessary information through multilayer operations, and to preserve important features, thus greatly improving the efficiency of training. Several networks designed by the ImageNet project have achieved state-of-the-art performance, including VGGNet, ResNet, and AlexNet [21–24]. Apart from these, more sophisticated networks have been designed to achieve excellent performance. Zachi et al developed a CNN with ten residual blocks and two fully connected layers to develop an AI-enabled ECG model. Each residual block uses two blocks to implement its functions, each of which consist of a batch-normalization layer, a nonlinear ReLU activation function, and a convolution layer. This model allows the analysis of ECGs during normal sinus rhythm to achieve atrial fibrillation screening and prediction with high AUC (0.90; 95% CL 0.90–0.91) [14]. Another CNN with six convolution layers, six maxpooling layers, and two fully connected layers was also developed by this team to identify individuals with a high risk of developing left ventricular dysfunction,achieving a high AUC of 0.93 [25]. The abovementioned resultsprovide a glimpse into the current state of CNN applications on ECG analysis. We used 2 commonly used models VGGNet16 and ResNet50, pretrained for image classification to perform transfer learning [24, 26]. The AUC values were 0.88, 0.87, and 0.76 for our CNN, and the VGGNet16, and ResNet50 models, respectively (S3 Fig). Our CNN performs better than the ResNet50, and has almost the same performance as the VGGNet16. Additionally, the convolutional neural network used in this study has a simpler structure, which can result in an increased working speed, and thusgreater computational efficiency. Moreover, the images we used were ECGs, of which only the waveform information needs to be analyzed, so it is unnecessary to analyze the color and other patterns of the picture, which would require a deeper network to extract enough information to achieve effective training. For example, images like histological slides may be input into a network as RGB images, while the ECGs, carrying less information, can be input as single-channel grayscale images. We thus developed a CNN model that is more suitable for analyzing ECG.

In our study, CNN achieved the highest accuracy among the three models (CNN, SVM, and RF). The nature of the CNN architecture explains its excellent performance. The CNN effectively implements multiple convolution operations, and imaging features become more abstract as the neural network depth increases, improving the accuracy of image classification

[27]. Additionally, the small number of preprocessing steps greatly alleviates the heavy burden that accompanies data extraction, and a more complete analysis of the image is possible.

We also analyzed the contributions of different variables in ML model development. HR was demonstrated to be the most relevant factor in the generation of the RF model. Existing studies have shown that HR serves as an important prognostic indicator in many cardiovascular diseases. One study showed that HR was a risk factor for predicting the mortality of patients with IS, and a low HR was an indicator of a better functional outcome [28]. In the SVM model, the QT interval was the most important variable, which represents repolarization disorder and increased susceptibility to cardiac arrhythmias [29]. Some studies have identified a prolonged QTc interval in 26% of patients without a primary heart disease history and suggested that it could serve as an important prognostic factor for long-term mortality [30, 31]. Additionally, the HR and QR intervals were significantly different between the N-N and N-IS datasets in the basic demographic analysis. All these findings are highly consistent with our study, adding credence to the feasibility of our models.

There are differences in the ECGs and basic ECG parameters between A-ISs and N-Ns. Similarly, the ECG scores calculated by the AI models were significantly between the N-Ns and A-Iss, as well as between the N-Ns and N-ISs, suggesting that the ECG scores can reflect difference between the images and their parameters. As shown in Fig 3D, the ECG scores of A-ISs were generally higher than the other datasets, suggesting that more abnormalities can be found in high-scoring ECGs. In addition, there were some high-scoring ECGs in the N-IS dataset, indicating that some of these abnormal patterns were also present in the N-ISs (Fig 3D), which suggests that routine approaches might overlook some concealed ECG changes. Scores over the optimal cutoff value set by AI models were considered toindicate abnormalities. In the N-IS dataset, 145 ECGs (21.55%) were identified to be abnormal by the CNN model, 54 (8.02%) by the RF model, and 84 (12.48%) by the SVM model. The CNN model identified the most abnormal ECGs in the N-IS dataset. Furthermore, the results for the validation cohort further confirmed our hypothesis. The generally normal ECGs of half of the post-IS patients (47.83%) were identified to be abnormal, further confirming that routine approaches might overlook some concealed ECG changes. Therefore, all post-IS patients with higher scores may require greater medical attention and close monitoring, despite potentially presenting with normal ECGs.

The mRS is used to assess the functional outcome of poststroke patients, but it is limited by its subjectivity and relatively poor reproducibility. Additionally, interobserver variability attenuates the reliability of outcome measures [32, 33]. Our study demonstrated that the ECG score assessed by the CNN model was correlated with the mRS score. We can obtain more accurate and objective results quickly with the help of AI. We collected the mRS scores of the post-IS patients when they were discharged from the hospital. The ECGs with higher mRS scores (poor prognosis) also had higher CNN scores, indicating that the CNN scores derived from our model can be used to predict the prognosis of post-IS patients. Thus, more follow-up and medical attention should be provided to patients with higher CNN scores in clinical practice.

To further investigate the abnormal regions identified by the CNN, we applied Grad-CAM to reconstruct the images and then captured and marked key abnormal regions. The feature extraction intensity was presented in a heatmap, and we found that feature visualization was most intense in the V4, V3, and V5 lead regions. The V4 lead region harbored the greatest number of key points, so it may be an important determinant in classifying abnormalities. Maximal T-wave inversion with widespread ST depression in V4 and V5 is a marker for assessing the prognosis of acute coronary syndromes [34]. Interestingly, some studies found acute coronary syndrome in 12.7% of patients after acute IS within 3 days [35]. An abnormal ECG may foreshadow subsequent acute coronary syndrome or reflect its presence, which also

explains the poor clinical outcome in some post-IS patients. These findings are in concordance with our results, further demonstrating the good performance of our models in extracting abnormal features from ECG in IS patients. Given various interfering factors, such as the distance between heart regions and electrodes, and some pathological conditions, measuring a single lead would increase sensitivity and specificity in diagnosis compared to fixed lead criteria [36, 37]. Lai et al hypothesized that optimal ECG-lead selection could reduce diagnostic redundancy and increase the generalizability of deep learning-based models, and they found that 4-lead ECG (leads II, aVR, V1, and V4) significantly outperformed 12-lead ECG in interpreting abnormalities [38]. Our study introduced a refined approach to ECG monitoring in post-IS patients, indicating that focusing on a subset of ECG leads (V3, V4, V5) may optimize abnormality detection and clinical outcome prognostication.

There are some limitations of our study. First, we did not include enough baseline characteristics in this study. Some biomarkers and measurements, such as blood pressure, cardiac troponin, and brain natriuretic peptide are important indicators for predicting the prognosis of IS [4, 39–43]. Second, more prospective research should be carried out to track the prognosis of post-IS patients to further confirm our study findings. Third, clinicians can use infarct size, a more reproducible measurement, to assess the prognosis of post-IS patients [44–46]. In this study, we used the mRS to predict the clinical outcome of post-IS patients, which is not as reproducible and significant as infarct size. More sophisticated algorithms will be developed in future studies. Additionally, we did not collect ECGs before the stroke attack, which could have induced confounding factors such as the natural variability among individuals. Finally, in this study, we observed changes in the ECGs after IS, but due to a lack of evidence, we cannot explain the causal relationship between cerebral infarction and ECG changes, only that there is a relationship between the two. In the future, the multiomics analysis combined with AI will be conducted to further investigate the inner molecular mechanisms of our findings.

## Conclusion

There are four main findings in our study. We found that a large proportion of post-IS ECGs (78.84%) had abnormal changes, most commonly T-wave changes. The three AI models we developed also uncovered abnormalities in these ECGs, including some ignored by routine diagnosis. The ECG scores assessed by the CNN model may be associated with prognosis in post-IS patients. Finally, the V4 lead may harbor the highest probability of abnormality. We successfully introduced an approach to assess the clinical outcomes and degree of disability in post-IS patients.

## Supporting information

**S1 Fig. Confusion matrixes of the three AI models.** The label on the left is the true label, and the label below is the predicted label. (A) The confusion matrix of the CNN model shows the absolute numbers of classifications made for patients belonging to the A-IS group and N-N group. (B) The confusion matrix of the RF model shows the absolute numbers of classifications made for patients in the A-IS and N-N groups. (C) The confusion matrix of the SVM model shows the absolute numbers of classifications made for patients in the A-IS and N-N groups. (TIF)

**S2 Fig. Score distribution of the N-N, N-IS, and A-IS datasets evaluated by the ML models.** (A) The score distribution of the N-N, N-IS and A-IS datasets was evaluated by the RF model. The scores between the N-N and N-IS datasets are significantly different ($p < 0.001$, Wilcoxon rank-sum test), as are the scores between the A-IS and N-IS datasets. (B) The score distribution

of the N-N, N-IS and A-IS datasets was evaluated by the SVM model. The scores between the N-N and N-IS datasets are significantly different ($p < 0.001$, Wilcoxon rank-sum test), as are the scores between the A-IS and N-IS datasets. ($p < 0.001$, Wilcoxon rank-sum test). Note: $p < 0.05$, *; $<0.01$, **; $<0.001$, ***.
(TIF)

**S3 Fig. ROC curves for Our CNN, VGGNet16, and ResNet50 models.**
(TIF)

**S1 Table. Network structure and hyperparameters of the CNN model.**
(DOCX)

**S2 Table. *P* values for the ECG parameters.**
(DOCX)

**S3 Table. Summary of AI model performance.**
(DOCX)

## Acknowledgments

The authors express their gratitude to American Journal Experts (https://www.aje.cn/) for the expert linguistic services provided.

## Author Contributions

**Conceptualization:** Ziqiang Zeng, Qixuan Wang, Libin Deng.

**Data curation:** Ziqiang Zeng, Yingjing Yu, Yichu Zhang, Qi Chen, Yuting Wang, Lingyu Yan, Zujue Cheng, Lijun Xu, Yingping Yi, Guangqin Fan, Libin Deng.

**Formal analysis:** Ziqiang Zeng, Qixuan Wang, Yingjing Yu, Yichu Zhang, Weiming Lou, Libin Deng.

**Funding acquisition:** Zujue Cheng, Lijun Xu, Yingping Yi.

**Investigation:** Libin Deng.

**Methodology:** Libin Deng.

**Supervision:** Libin Deng.

**Validation:** Libin Deng.

**Writing – original draft:** Qixuan Wang.

**Writing – review & editing:** Ziqiang Zeng, Qixuan Wang, Libin Deng.

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
