## [Decision Letter · Decision Letter 0]

6 Nov 2022

PONE-D-22-27898Assessing electrocardiogram changes after ischemic stroke with artificial intelligencePLOS ONE

Dear Dr. Deng,

Thank you for submitting your manuscript to PLOS ONE. After careful consideration, we feel that it has merit but does not fully meet PLOS ONE’s publication criteria as it currently stands. Therefore, we invite you to submit a revised version of the manuscript that addresses the points raised during the review process.

We look forward to receiving your revised manuscript.

Kind regards,

Felix Albu, Ph.D.

Academic Editor

PLOS ONE

Journal Requirements:

3. Ethics statement appears in the Methods section of the manuscript AND at the end of the manuscript:

Your ethics statement should only appear in the Methods section of your manuscript. If your ethics statement is written in any section besides the Methods, please delete it from any other section.

Additional Editor Comments:

The decision is Major Revision.

Reviewers' comments:

Reviewer's Responses to Questions

**Comments to the Author**

1. Is the manuscript technically sound, and do the data support the conclusions?

Reviewer #1: Yes

Reviewer #2: Yes

2. Has the statistical analysis been performed appropriately and rigorously? 

Reviewer #1: No

Reviewer #2: Yes

3. Have the authors made all data underlying the findings in their manuscript fully available?

Reviewer #1: No

Reviewer #2: Yes

4. Is the manuscript presented in an intelligible fashion and written in standard English?

Reviewer #1: Yes

Reviewer #2: Yes

5. Review Comments to the Author

Reviewer #1: The contribution is limited given the fact that the proposed work was not compared against state of the art deep-learning methods.

In my opinion, a fair literature review should be provided in a separate section; then, a comparison can be made.

upon completing this action, the contribution can be fairly jusdged.

Reviewer #2: The following changes should be done to improve the manuscript

1. No methods have mentioned for extraction of ECG parameter. Kindly, mention the methods which have been used to extract the ECG parameters.

2. Performance comparison table is missing, kindly add it.

3. Why only SVM is used in machine learning technique should be clearly mentioned.

6. PLOS authors have the option to publish the peer review history of their article (what does this mean?). If published, this will include your full peer review and any attached files.

Reviewer #1: **Yes: **Ibrahim Sadek

Reviewer #2: No

---

## [Author Response · Author response to Decision Letter 0]

13 Nov 2022

Dear Editors:

Thank you for your letter comments concerning our manuscript entitled “Assessing electrocardiogram changes after ischemic stroke with artificial intelligence” (PONE-D-22-27898). Those comments are all valuable and very helpful for revising and improving our paper, as well as the important guiding significance to our research. We have studied the comments carefully and made corrections, which we hope meet with approval. Revised portions are marked in red or blue on the paper. The main corrections in the paper and the responses to the editors are as follows:

Responds to the Journal’s requirements:

(1) Requirements: Please ensure that your manuscript meets PLOS ONE's style requirements, including those for file naming.

Response: Thanks for mentioning the problems. We have checked and modified the article's format according to the magazine's requirements.

(2) Requirements: Please upload your study’s minimal underlying data set as either Supporting Information files or to a stable, public repository and include the relevant URLs, DOIs, or accession numbers within your revised cover letter.

Response: Thanks for pointing out this problem. We have explained the original data structure in detail in the revised cover letter. All original data we used is available now in the figshare database (https://doi.org/10.6084/m9.figshare.21523527.v1; DOI:10.6084/m9.figshare.21523527;) for reproduction.

(3) Requirements: Your ethics statement should only appear in the Methods section of your manuscript. If your ethics statement is written in any section besides the Methods, please delete it from any other section.

Response: Thanks for your reminder. We have moved the ethics statement from the end of the article to the materials and method section.

Responses to the reviewers’ comments:

Reviewer #1: The contribution is limited given the fact that the proposed work was not compared against state of the art deep-learning methods. In my opinion, a fair literature review should be provided in a separate section; then, a comparison can be made. upon completing this action, the contribution can be fairly jusdged.

Response: Thank you for your advice. We have added some literature review on the current state of CNN applications on ECG analysis in a separate paragraph in the discussion. Also, we input the same data into the Resnet50 network and VGG-16 network (the widely used developed CNN networks) to perform transfer learning, and compared the performances of these two networks with the network used in this study. The results show that there is no significant difference in the performance of the three networks. Also, the convolutional neural network we developed has a simpler structure, which can improve the running speed of the model, to improve the computing efficiency. At the same time, the images we used are ECGs, of which the waveform information needs to be analyzed, so it is unnecessary to analyze the color and other patterns of the picture which requires more deep network to extract enough information to achieve effective training. For example, images like histological slides are input into the network as RGB images which contains a lot of color information, while the electrocardiogram images are input as waveform grayscale images, carrying less information. We developed a CNN model that is more suitable to analysis ECG. We have added this in the discussion section.

The result of added transfer learning ( ROC for comparing the performance of CNN, VGGNet16, and ResNet50 ) has been uploaded as S3 Fig in supporting information.

Reviewer #2:

(1). No methods have mentioned for extraction of ECG parameter. Kindly, mention the methods which have been used to extract the ECG parameters.

Response: Thank you for your helpful comments. The ECG parameters were generated by the FX-7042 Cardimax ECG machine and recorded in the electronic medical record. Seven ECG parameters, including heart rate, P wave, P-R interval, QRS wave, QT interval, QTc interval, and RV5+SV1 value, were extracted from the electronic medical record and checked manually. We described it in the Material and method section.

(2). Performance comparison table is missing, kindly add it.

Response: Thank you for your helpful comments. The performance comparison table is in the supporting information (S3 Table. Summary of AI model performance)

(3). Why only SVM is used in machine learning technique should be clearly mentioned.

Response: Thank you for your helpful comments. In this paper, we have used two machine learning models, the support vector machine model, the random forest model, and one CNN model. CNN performs the best among the three. Unlike CNN which inputs images directly, the machine learning models we built used extracted ECG parameters to perform model training and classification. The weight of each parameter in model development can also be shown as importance, achieving model interpretation to some extent that which parameters are the most important indicators. We have improved this description in the method section.

We tried our best to improve the manuscript and made some changes in the manuscript. These changes will not influence the content and framework of the paper. And here we did not list the changes but marked them in red in the revised paper.

 We appreciate for Editors/Reviewers’ warm work sincerely and hope that the correction will meet with approval. Once again, thank you very much for your comments and suggestions.

Sincerely yours,

Libin Deng

Professor 

School of Public Health, Nanchang University; Jiangxi Provincial Key Laboratory of Preventive Medicine, Nanchang University; The Institute of Periodontal Disease, Nanchang University

No.461, BaYi Road, Nanchang, Jiangxi Province, P.R.C

Tel: +86-15170401580 

E-mail: lbdeng@ncu.edu.cn

---

## [Decision Letter · Decision Letter 1]

24 Nov 2022

PONE-D-22-27898R1Assessing electrocardiogram changes after ischemic stroke with artificial intelligencePLOS ONE

Dear Dr. Deng,

Thank you for submitting your manuscript to PLOS ONE. After careful consideration, we feel that it has merit but does not fully meet PLOS ONE’s publication criteria as it currently stands. Therefore, we invite you to submit a revised version of the manuscript that addresses the points raised during the review process. Please submit your revised manuscript by Jan 08 2023 11:59PM. If you will need more time than this to complete your revisions, please reply to this message or contact the journal office at plosone@plos.org. Please include the following items when submitting your revised manuscript:A rebuttal letter that responds to each point raised by the academic editor and reviewer(s). You should upload this letter as a separate file labeled 'Response to Reviewers'.A marked-up copy of your manuscript that highlights changes made to the original version. You should upload this as a separate file labeled 'Revised Manuscript with Track Changes'.An unmarked version of your revised paper without tracked changes. You should upload this as a separate file labeled 'Manuscript'.

We look forward to receiving your revised manuscript.

Kind regards,

Felix Albu, Ph.D.

Academic Editor

PLOS ONE

Journal Requirements:

Additional Editor Comments (if provided):

The decision is Minor Revision.

Reviewers' comments:

Reviewer's Responses to Questions

**Comments to the Author**

1. If the authors have adequately addressed your comments raised in a previous round of review and you feel that this manuscript is now acceptable for publication, you may indicate that here to bypass the “Comments to the Author” section, enter your conflict of interest statement in the “Confidential to Editor” section, and submit your "Accept" recommendation.

Reviewer #1: All comments have been addressed

Reviewer #2: (No Response)

2. Is the manuscript technically sound, and do the data support the conclusions?

Reviewer #1: Partly

Reviewer #2: Yes

3. Has the statistical analysis been performed appropriately and rigorously? 

Reviewer #1: Yes

Reviewer #2: Yes

4. Have the authors made all data underlying the findings in their manuscript fully available?

Reviewer #1: Yes

Reviewer #2: Yes

5. Is the manuscript presented in an intelligible fashion and written in standard English?

Reviewer #1: No

Reviewer #2: Yes

6. Review Comments to the Author

Reviewer #1: All comments have been addressed for the manuscript (Assessing electrocardiogram changes after ischemic stroke with artificial intelligence).

Reviewer #2: The following points to be addressed in the revised manuscript:

1. Abstract should be rewritten, as the work is not properly reflected in the abstract.

2. In 153 number line authors have written, they have proposed a CNN model. But in presentation of content, it does not look like. It should be changed to they have used CNN model.

3. They have used certain parameters for CNN model, the reason for use of particular parameter has not mentioned. Kindly mention it.

4. Authors have written about AuC and ROC, diagram has not given.

7. PLOS authors have the option to publish the peer review history of their article (what does this mean?). If published, this will include your full peer review and any attached files.

Reviewer #1: **Yes: **Ibrahim Sadek

Reviewer #2: No

---

## [Author Response · Author response to Decision Letter 1]

4 Dec 2022

Dear Editors:

Thank you for your letter comments concerning our manuscript entitled “Assessing electrocardiogram changes after ischemic stroke with artificial intelligence” (PONE-D-22-27898). Those comments are all valuable and very helpful for revising and improving our paper, as well as the important guiding significance to our research. We have studied the comments carefully and made corrections, which we hope meet with approval. Revised portions are marked on the paper. The main corrections in the paper and the responses to the editors are as follows:

Responds to the Journal’s requirements:

(1) Requirements: Please review your reference list to ensure that it is complete and correct. If you have cited papers that have been retracted, please include the rationale for doing so in the manuscript text, or remove these references and replace them with relevant current references. Any changes to the reference list should be mentioned in the rebuttal letter that accompanies your revised manuscript. If you need to cite a retracted article, indicate the article’s retracted status in the References list and also include a citation and full reference for the retraction notice.

Response: Thank you for the requirement. We have checked and edited the reference, making sure it meets the criteria of the journal. The details are shown shown in the revised manuscript.

Responses to the reviewers’ comments:

Reviewer #1: All comments have been addressed for the manuscript (Assessing electrocardiogram changes after ischemic stroke with artificial intelligence).

Response: Thank you for your guidance, we are very glad to revise the article under your guidance.

Reviewer #2:

(1). Abstract should be rewritten, as the work is not properly reflected in the abstract.

Response: Thank you for your advice, we have rewritten the abstract.

(2). In 153 number line authors have written, they have proposed a CNN model. But in presentation of content, it does not look like. It should be changed to they have used CNN model.

Response: Thank you for your comments, we have checked and revised this in the manuscript.

(3). They have used certain parameters for CNN model, the reason for use of particular parameter has not mentioned. Kindly mention it.

Response: Thanks for pointing out this problem. We adjusted the hyperparameters and structure of the convolutional neural network according to the working speed and performance of the model, based on which we chose the optimal hyperparameters and structure for the model. The details are shown in Method section. 

(4). Authors have written about AuC and ROC, diagram has not given.

Response: Thank you for pointing out this. The figures 1-5 are in the main body of the manuscript. S1-S3 figures are shown in supplemental material. The AUC and ROC diagrams are Figure 3A in the main body and S3 figure in the supplemental material. Here are the titles of the figures.

① Fig 1. Pipeline of AI model development and CNN structure. 

② Fig 2. Analysis of participant ECG parameters. 

③ Fig 3. Performance of the ML and CNN models.

④ Fig 4. CNN model performance in the validation cohort and in poststroke disability assessment.

⑤ Fig 5. Grad-CAM feature visualization.

⑥ S1 Fig. Confusion matrixes of the three AI models.

⑦ S2 Fig. Score distribution of the N-N, N-IS, and A-IS datasets evaluated by the ML models. 

⑧ S3 Fig. ROC curves for Our CNN, VGGNet16, and ResNet50 models.

We tried our best to improve the manuscript and made some changes in the manuscript. These changes will not influence the content and framework of the paper. And here we did not list the changes but marked them in red in the revised paper.

 We appreciate for Editors/Reviewers’ warm work sincerely and hope that the correction will meet with approval. Once again, thank you very much for your comments and suggestions.

Sincerely yours,

Libin Deng

Professor 

School of Public Health, Nanchang University; Jiangxi Provincial Key Laboratory of Preventive Medicine, Nanchang University; The Institute of Periodontal Disease, Nanchang University

No.461, BaYi Road, Nanchang, Jiangxi Province, P.R.C

Tel: +86-15170401580 

E-mail: lbdeng@ncu.edu.cn

---

## [Decision Letter · Decision Letter 2]

13 Dec 2022

Assessing electrocardiogram changes after ischemic stroke with artificial intelligence

PONE-D-22-27898R2

Dear Dr. Deng,

We’re pleased to inform you that your manuscript has been judged scientifically suitable for publication and will be formally accepted for publication once it meets all outstanding technical requirements.

Kind regards,

Felix Albu, Ph.D.

Academic Editor

PLOS ONE

Additional Editor Comments (optional):

The decision is Accept.

Reviewers' comments:

Reviewer's Responses to Questions

**Comments to the Author**

1. If the authors have adequately addressed your comments raised in a previous round of review and you feel that this manuscript is now acceptable for publication, you may indicate that here to bypass the “Comments to the Author” section, enter your conflict of interest statement in the “Confidential to Editor” section, and submit your "Accept" recommendation.

Reviewer #1: All comments have been addressed

Reviewer #2: All comments have been addressed

2. Is the manuscript technically sound, and do the data support the conclusions?

Reviewer #1: Yes

Reviewer #2: Yes

3. Has the statistical analysis been performed appropriately and rigorously? 

Reviewer #1: Yes

Reviewer #2: Yes

4. Have the authors made all data underlying the findings in their manuscript fully available?

Reviewer #1: Yes

Reviewer #2: Yes

5. Is the manuscript presented in an intelligible fashion and written in standard English?

Reviewer #1: No

Reviewer #2: Yes

6. Review Comments to the Author

Reviewer #1: All comments have been addressed for the manuscript (Assessing electrocardiogram changes after ischemic stroke with artificial intelligence).

Reviewer #2: All comments have been addressed for the manuscript (Assessing electrocardiogram changes after ischemic stroke with artificial intelligence).

7. PLOS authors have the option to publish the peer review history of their article (what does this mean?). If published, this will include your full peer review and any attached files.

Reviewer #1: **Yes: **Ibrahim Sadek

Reviewer #2: No

---

## [Editor Report · Acceptance letter]

16 Dec 2022

PONE-D-22-27898R2 

Assessing electrocardiogram changes after ischemic stroke with artificial intelligence 

Dear Dr. Deng:

I'm pleased to inform you that your manuscript has been deemed suitable for publication in PLOS ONE. Congratulations! Your manuscript is now with our production department. 

Kind regards, 

on behalf of

Dr. Felix Albu 

Academic Editor

PLOS ONE